# Case Report: Successful Treatment of Solitary Fibrous Tumor with Selective Internal Radiation Therapy (SIRT)

**DOI:** 10.3390/diseases12110290

**Published:** 2024-11-12

**Authors:** Omar Badran, Sergey Dereza, Labib Mireb, Ziv Neeman, Gil Bar-Sela

**Affiliations:** 1Department of Oncology, Emek Medical Center, Yitzhak Rabin Boulevard 21, Afula 1834111, Israel; gil_ba@clalit.org.il; 2Technion Integrated Cancer Center, Faculty of Medicine, Technion, Haifa 1834111, Israel; ziv_ne@clalit.org.il; 3Diagnostic Imaging Department, Emek Medical Center, Yitzhak Rabin Boulevard 21, Afula 1834111, Israel; sergey_de@clalit.org.il (S.D.); miram_la@clalit.org.il (L.M.)

**Keywords:** solitary fibrous tumor (SFT), selective internal radiation therapy (SIRT), hypervascular tumor, Yttrium-90 microspheres

## Abstract

**Background**: This case report details the innovative use of selective internal radiation therapy (SIRT) with Yttrium-90 resin microspheres to treat a 73-year-old woman with a solitary fibrous tumor (SFT), a rare and challenging tumor type. SFTs often present significant treatment difficulties, especially in cases of recurrence or metastasis, as systemic therapies typically show limited effectiveness. This report explores SIRT as an alternative therapeutic approach for SFTs with liver metastasis. **Methods**: The patient initially presented with a pelvic mass, which was surgically resected. However, metastatic disease later developed in the liver. After experiencing severe side effects from targeted therapy with sunitinib, the patient was selected for treatment with SIRT as an alternative. **Results**: Following the SIRT intervention, the patient demonstrated a substantial reduction in tumor size and significant relief from symptoms. This outcome suggests SIRT’s effectiveness as a targeted treatment for metastatic SFT. **Conclusions**: To our knowledge, and based on an extensive literature review, this is the first reported instance of treating SFT with SIRT. This case provides new insights into SIRT’s potential as a therapeutic strategy, particularly for patients for whom conventional treatments are either ineffective or intolerable. The success observed here underscores SIRT’s potential as a less invasive, locally targeted treatment option, offering hope for similar cases.

## 1. Introduction

Solitary fibrous tumor (SFT) is a rare mesenchymal tumor that accounts for less than 2% of all soft tissue tumors [1]. These tumors predominantly affect middle-aged adults, typically between the ages of 20 and 70, and present equally in both men and women [2,3]. While initially described in the pleura, SFTs can occur in nearly any location, most commonly affecting the pleura, accounting for about 30% of cases [4]. However, these tumors are also found in other places, such as the meninges, abdominal cavity, trunk, and extremities. Intracranial SFTs are particularly rare, representing a small fraction of meningeal and intracranial tumors [5]. These tumors are often asymptomatic until they grow large enough to cause a mass effect on adjacent organs, leading to symptoms [6].

On contrast-enhanced computed tomography (CT) scans, SFTs typically appear as well-circumscribed, hypervascular tumors, often presenting as lobulated masses with necrosis, especially in more extensive tumors [7]. Approximately 65% of cases exhibit avid contrast enhancement in the arterial and early portal venous phases, with contrast washout in the delayed phase, particularly if fibrous components predominate [8,9]. Some tumors also show large collateral feeding vessels, which may aid in diagnosing more aggressive forms [8,9]. Heterogeneous enhancement, seen in up to 76.5% of aggressive SFT cases, reflects the presence of necrosis, hemorrhage, or cystic changes [8,9]. These radiologic features, including large feeding vessels and heterogeneity in enhancement, are critical in assessing the tumor’s aggressiveness and guiding treatment decisions, especially in more challenging anatomical locations like the abdomen or retroperitoneum [9]. SFTs display a mix of solid and cystic components on magnetic resonance imaging (MRI) [9]. Solid areas are typically isointense to hypointense relative to skeletal muscle on both T1- and T2-weighted images, while cystic regions appear hyperintense [10]. These tumors show substantial enhancement after contrast administration, reflecting their vascularity, similar to CT imaging, and MRI is functional for identifying necrosis, hemorrhage, or cystic changes [10]. While distinguishing between indolent and aggressive SFTs is limited, 18F-fluorodeoxyglucose positron emission tomography (FDG-PET/CT) can detect metastasis and recurrence, as malignant SFTs tend to show higher FDG uptake [11]. It may also aid in monitoring treatment response, helping to assess disease progression and guide further management.

Histologically, SFTs are characterized by spindle-to-ovoid cells in a patternless distribution, with dense cellular areas alternating with hypocellular regions rich in stromal collagen [12]. Immunohistochemical staining, particularly for markers such as CD99, CD34, and BCL2, assists in differentiating SFT from other spindle cell tumors. Molecular genetic analysis is crucial in confirming the diagnosis, particularly identifying the NAB2-STAT6 gene fusion, which is highly specific and sensitive for SFT [13,14]. However, the differential diagnosis should include other soft tissue tumors like dedifferentiated liposarcoma, which may show overlapping features [15].

Management of SFT typically involves a multidisciplinary approach, with surgical resection being the primary treatment modality aimed at achieving wide margins to reduce recurrence risk [16]. Recurrence rates for SFTs vary, with estimates ranging from 10% to 20%, but some studies with more extended follow-up periods have reported recurrence rates exceeding 30% [17,18]. Factors such as the mitotic index, necrosis, KI67 index, and telomerase reverse transcriptase (TERT) promoter mutations have been identified as significant predictors of recurrence [19]. Therefore, risk stratification using systems such as the modified Demicco score is crucial for guiding post-surgical management [19]. For patients who exhibit high-risk features, such as positive surgical margins or high mitotic count, adjuvant radiotherapy (RT) may reduce the risk of local recurrence [20]. However, no definitive overall survival benefit has been demonstrated in observational studies [20].

Systemic therapies for advanced SFT have been explored, but the rarity of the disease limits large-scale studies. Traditional chemotherapy has shown limited efficacy, with low response rates [21]. Anthracycline-based chemotherapy generally yields a modest response, with only a tiny proportion of patients showing tumor shrinkage, while a larger group achieves disease stabilization [22]. Progression-free survival with this treatment typically lasts a few months, and overall survival averages around one year [23]. Ifosfamide, another chemotherapy option, has limited efficacy, producing tumor responses in a small percentage of patients [23]. Dacarbazine, however, tends to perform better, offering a higher likelihood of tumor response and slightly longer progression-free intervals [23]. While less effective at shrinking tumors, Trabectedin is notable for its ability to stabilize disease in many cases, providing patients with a meaningful extension of disease control and overall survival [23].

Since solitary fibrous tumors (SFTs) are highly vascularized, with the overexpression of angiogenic pathways such as platelet-derived growth factor receptor (PDGFR) and vascular endothelial growth factor receptor (VEGFR) [23,24], targeted therapies like sunitinib, sorafenib, and pazopanib have shown some potential [23]. Sunitinib has demonstrated a modest effect, offering partial tumor shrinkage and stabilization in about half of the patients treated [24]. On the other hand, pazopanib appears to be more effective, particularly in typical SFTs, where a significant number of patients experience either tumor reduction or disease control [25]. However, the benefits in more aggressive tumor forms, such as malignant or dedifferentiated SFTs, are somewhat reduced [25]. Despite the potential of these therapies, severe side effects can limit their use, as seen in our own experience, where treatment discontinuation was necessary in some cases [23,24,25].

Given these limitations, there is a growing interest in alternative treatment modalities that can provide effective tumor control while minimizing systemic toxicity. Since this is a case of liver-only metastatic disease with hypervascular lesions, the option for selective internal radiation therapy (SIRT) was considered. SIRT is a minimally invasive treatment used for primary and secondary liver cancers, where millions of tiny radioactive resin-based particles, known as SIR-spheres, are delivered intra-arterially via feeding arteries into liver tumors [26]. These particles release beta radiation, mainly targeting and destroying tumor cells while relatively sparing surrounding healthy tissue [26]. SIRT has proven to be particularly effective for treating inoperable liver tumors, including primary liver cancers such as hepatocellular carcinoma (HCC) and metastatic liver tumors from colorectal cancer and neuroendocrine tumors [27]. Clinical studies have shown that SIRT can enhance response rates, improve local tumor control, and, in some cases, extend both progression-free survival (PFS) and overall survival (OS) [28].

Historically, external beam radiation therapy (EBRT) has been widely used for treating unresectable liver metastases [29]. While techniques like stereotactic body radiation therapy (SBRT) have shown promising results in achieving local control, the treatment poses challenges [30]. Managing respiratory motion, ensuring precision with fiducial markers, and dealing with increased radiation exposure to normal liver tissue are all significant concerns [30,31]. Moreover, multiple treatment sessions are often required, and there are limitations in delivering sufficient radiation doses to larger tumors, making this approach less ideal in some cases [31].

In contrast, selective internal radiation therapy (SIRT) offers a more targeted approach, delivering higher doses of radiation directly to the tumor through the liver’s arterial supply while sparing surrounding healthy tissue [32]. Studies in patients with non-resectable liver metastases from colorectal cancer have shown that SIRT, when combined with chemotherapy, can enhance tumor response and delay disease progression in the liver [33]. SIRT was also tested in combination with sorafenib in patients with advanced hepatocellular carcinoma. The study found no significant improvement in overall survival when SIRT was added to sorafenib compared to using sorafenib alone; however, some subgroups, such as patients without cirrhosis, those with non-alcoholic cirrhosis, and patients under 65, showed potential survival benefits [34].

However, despite its targeted nature, SIRT carries risks, including the potential for unintended radiation exposure to nearby organs like the stomach, pancreas, or lungs, which can result in complications such as pancreatitis or radiation pneumonitis [35]. Careful patient selection, detailed diagnostic evaluations, and precise treatment planning using Tc-99m macroaggregated albumin (Tc-99 MMA) for simulation, followed by accurate delivery of SIR-spheres, are crucial to minimizing these risks and achieving the best possible outcomes [36].

In this case, we present the novel application of SIRT for treating a metastatic SFT to the liver. We demonstrate significant tumor regression and symptom relief, suggesting that SIRT could provide a viable alternative for SFT patients who are not suitable for surgery or systemic therapy.

SIRT presents a promising avenue for managing SFT and potentially other rare tumors by offering a targeted treatment option with a favorable safety profile. The positive outcome observed in this patient underscores the potential of SIRT to expand the therapeutic landscape for SFT, providing new hope for patients with limited options. Further research is needed to establish the role of SIRT in SFT management and to determine its efficacy and safety in a broader patient population.

## 2. Case Presentation

A 73-year-old woman, married with two children and a retired dentist, presented with a medical history notable for hypertension and spinal stenosis at L4-S1. She had undergone a laminectomy in 2017, which did not alleviate her back pain or mobility issues. She had no family history of cancer.

In January 2017, the patient was evaluated for a pelvic mass initially suspected to be an ovarian cyst. She underwent an oophorectomy, during which a highly vascular retroperitoneal para-rectal mass measuring 5 cm was discovered, located to the left of the uterus. A colonoscopy that was performed that year was unremarkable. In March 2017, she underwent surgery to resect the mass. The histopathological analysis identified the mass as a hemangiopericytoma, which had infiltrated surrounding fat, with positive surgical margins and a high mitotic index.

Further pathological review reclassified the tumor as an SFT, with CD99- and STAT6-positive and CD34-negative, KI67-7%, measuring 7 cm. The patient subsequently received adjuvant radiation therapy to the pelvis, totaling 5040 cGy, completed by June 2017. She was placed under regular follow-up after this treatment.

In the years following her treatment, the patient remained under surveillance. An MRI of the abdomen and pelvis in August 2022 showed no signs of recurrence, and a chest, stomach, and pelvis CT scan in July 2022 was also normal. Additionally, a colonoscopy performed in June 2022 did not reveal any abnormalities. However, an MRI conducted in June 2023 showed several new small hypervascular liver lesions, the largest of which had been seen on previous imaging but was not detected in earlier studies. These findings were initially interpreted as focal nodular hyperplasia (FNH), but a subsequent review by a radiologist raised the suspicion of hypervascular metastases.

A biopsy of one of the liver lesions performed in July 2023 confirmed the presence of metastatic SFT. Genetic testing of the tumor revealed a low mutation burden, microsatellite instability (MSI) stable status, and a TERT C.124c>T mutation, which has prognostic significance but no direct therapeutic implication. After discussing the treatment options with the patient, which included chemotherapy or TKI therapy, and explaining the benefits, advantages, and disadvantages of each treatment, it was agreed to start therapy with sunitinib.

The patient began treatment with sunitinib at 37.5 mg daily in September 2023. However, after one week, she experienced severe weakness and nausea, leading to the discontinuation of the therapy. The treatment was restarted at a reduced dose of 25 mg daily in October, which was better tolerated. Unfortunately, by mid-November, the patient developed severe side effects, including ulcers on her heels and inguinal area, vaginal burning, joint pain, and peeling of the skin on her hands, consistent with sunitinib toxicity. As a result, the treatment was halted, and she received local supportive care.

In December 2023, a full-body CT scan revealed that two known liver metastases had grown, and a new lesion had appeared near the inferior vena cava (IVC) in the liver. A multi-phase liver CT protocol performed later in December identified five lesions in the right liver lobe and at least one lesion in the left lobe (Figure 1). After consultation with an interventional radiologist, it was determined that the disease spread pattern was unsuitable for chemoembolization. Still, it was appropriate for selective internal radiation therapy (SIRT) using Yttrium-90. In February 2024, approximately one week before SIRT treatment, the patient underwent preparatory hepatic catheterization and selective simulation using Tc-99mma, demonstrating the feeding arteries to the liver tumors, and a low lung shunt fraction of 9% was calculated. During this preparatory hepatic catheterization simulation, the arteries feeding the tumors were mapped, allowing optimal catheter location placement (Figure 2). On 8 February 2024, the SIRT procedure was conducted under general anesthesia due to the patient’s back pain and inability to lie flat; a puncture was performed under local anesthesia, with fluoroscopy and ultrasound guidance, in the right common femoral artery. Catheterization of the superior mesenteric artery (SMA) was performed, and pressure injection showed no vessels from the SMA supplying the tumor. Subsequently, catheterization of the celiac artery was performed, with automatic injection showing celiac artery anatomy and pathological tumor enhancement in both liver lobes, particularly in the right lobe. A branch from the common hepatic artery was identified, supplying part of the right liver lobe, including at least two tumors. An accessory branch to the left lobe was observed through the left gastric artery, though it did not supply the liver tumors. Three doses of technetium were injected selectively into the right hepatic artery distal to the cystic artery, the left hepatic artery and a branch supplying part of the tumors in the right lobe, which originates from the proximal common hepatic artery. Due to the proximity of the origin of the right gastric artery, embolization with 2 mm coils was performed. At the end of the scan, the patient was transferred to the Nuclear Medicine Department, which showed a minimal lung shunt (9%) with no evidence of gastrointestinal shunt. The patient was then returned to the angiography suite for the second phase of the procedure. Selective catheterization of the three arteries was repeated, and technetium was injected with a total calculated dose of 1.4 GBq (0.4 to the left lobe, 0.4 to the systolic artery in the right lobe, and 0.5 to the right hepatic artery distal to the cystic artery). SIRT treatment was performed successfully with no apparent complications, and the patient was discharged home the next day.

A post-SIRT treatment follow-up PET-CT scan in April 2024 showed no pathological uptake. The previously identified hypervascular liver lesions were absent, and new, ill-defined hypodense changes were observed in both liver lobes, primarily on the right, potentially representing post-SIRT changes (Figure 3). The interventional radiologist and diagnostic radiologist team reviewed the findings. They confirmed that these changes were consistent with expected post-treatment effects, indicating an excellent response, with the lesions now appearing as cystic cavities.

## 3. Discussion

This case illustrates the potential of SIRT as a treatment option for SFT, especially in scenarios where traditional options such as surgery or systemic therapy are not feasible. Although rare, SFTs can pose significant treatment challenges due to their rarity, sluggish behavior, lack of sufficient research, and the limited effectiveness of available systemic therapies [1]. The hypervascular nature of SFTs makes them particularly suitable for SIRT treatment, as this approach leverages the physiologic tumor arterial-based blood supply to deliver high-dose, localized radiation primarily to the tumors while sparing the normal liver tissue, which gains its blood supply mainly from the portal vein [7]. Unlike external beam radiation therapy, which may have limited dosing and precision [30,31], SIRT allows for administering significantly higher radiation doses specifically to the tumor, enhancing the therapeutic effect while minimizing exposure and damage to surrounding healthy tissues [32]. This targeted delivery is especially advantageous in managing hypervascular tumors like SFT, where achieving sufficient radiation doses through traditional methods can be challenging [37]. Performing SIRT as an outpatient procedure offers several advantages, such as improving patient convenience, minimizing hospitalization time, reducing the risk of complications, and lowering overall costs [37]. Furthermore, the option to complete the treatment in a single day could make it particularly attractive for patients who are either unsuitable or hesitant to undergo more invasive procedures, thereby expanding its utility in managing tumors like SFT [37].

In this patient, SIRT led to significant tumor regression lesions, suggesting that it could be a valuable tool in managing liver-limited SFTs with those that are not suitable for other treatments. The ability of SIRT to deliver targeted radiation while minimizing systemic exposure is especially beneficial for patients who cannot tolerate the side effects of systemic therapies, such as this case, which caused severe toxicity and required discontinuation.

While the positive outcome in this case is encouraging, it is essential to approach these findings cautiously. Further research is needed to explore the broader applicability of SIRT in SFT, including well-designed clinical trials to assess its safety, efficacy, and potential role in the treatment algorithm for this rare tumor.

Given the rarity of SFT and the limited treatment options, it would be valuable to investigate SIRT in a basket trial that includes other rare hypervascular tumors. Such a trial could help define the subsets of patients who may benefit most from this therapy based on tumor biology and disease characteristics. This approach would provide more robust data on the efficacy of SIRT in SFT and contribute to understanding its role in other rare malignancies.

## 4. Conclusions

The successful management of SFT with SIRT in this case suggests that it could be a valuable treatment strategy for similar cases, particularly for patients with contraindications to conventional therapies. Further studies are needed to evaluate SIRT’s broader applicability in SFT management.

## Figures and Tables

**Figure 1 diseases-12-00290-f001:**
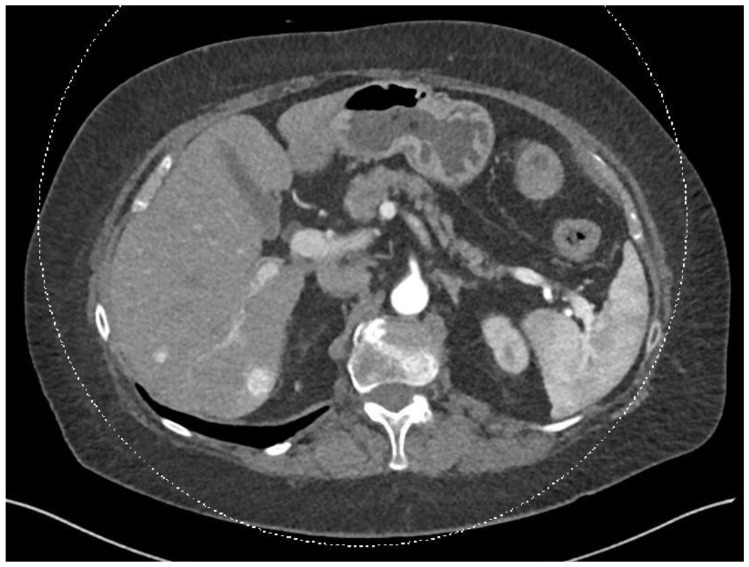
This abdominal and pelvic CT with contrast, performed on 25 December 2023, was compared to the previous abdominal CT from 15 August 2023 and the chest CT from 9 December 2021. The liver demonstrated two hypervascular lesions consistent with SFT: one lesion in segment 8 measuring 0.9 cm, previously 0.6 cm, and another lesion in segment 6 measuring 1.5 cm, previously 1.3 cm.

**Figure 2 diseases-12-00290-f002:**
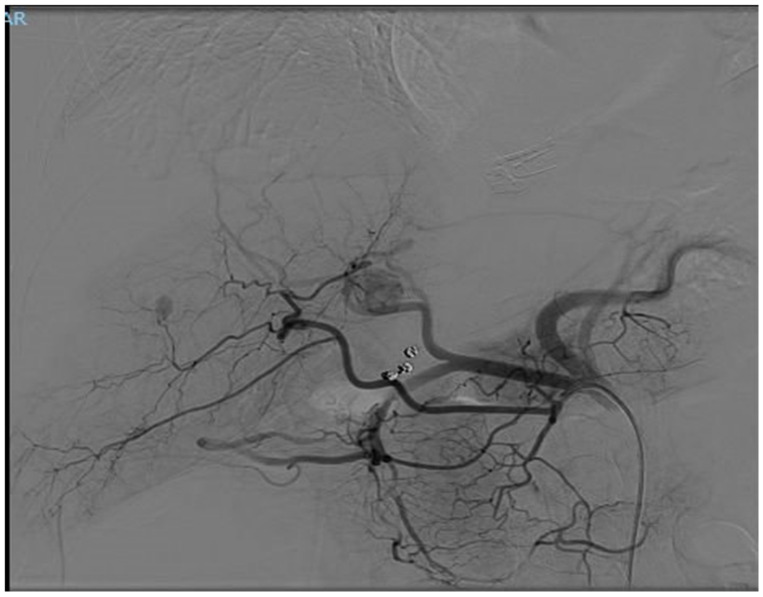
This CT angiography, performed on 8 February 2024, simulates contrast injection, highlighting the blood supply to the tumors during the arterial phase.

**Figure 3 diseases-12-00290-f003:**
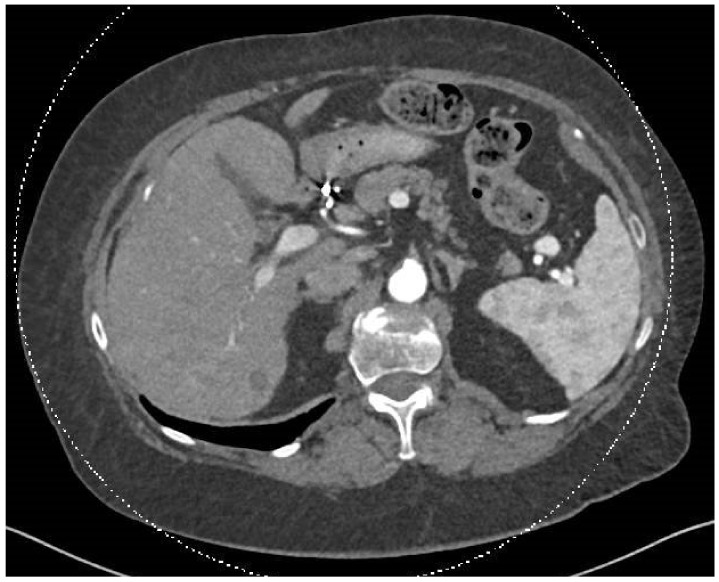
In this CT scan, dated 30 July 2024, of the chest, abdomen, and pelvis, performed after intravenous contrast injection with additional oral contrast administration, pre- and post-contrast imaging was conducted using a triphasic protocol focused on the liver. Comparison was made with the previous CT examination from 15 May 2024. The smaller lesions disappeared following SIRT treatment, and the dominant lesion has become hypodense, indicating an inactive metastasis.

## Data Availability

The original contributions presented in the study are included in the article; further inquiries can be directed to the corresponding author.

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
