# Peer review of "Case Report: Successful Treatment of Solitary Fibrous Tumor with Selective Internal Radiation Therapy (SIRT)"

_diseases, 2024, doi:10.3390/diseases12110290_

Round 1
Reviewer 1 Report
Comments and Suggestions for Authors
In the paper entitled “Case Report: Successful Treatment of Solitary Fibrous Tumor 1 with Selective Internal Radiation Therapy (SIRT)”, the authors Badran O, Dereza S, Labib M, Ziv N and Bar-Sela G described a case report of a successful treatment of a patient with Solitary Fibrous Tumor by SIRT approach, apparently first such a case described in literature.
Considering the seriousness of SFT and lack of options to treat it successfully, this case gives the hope that SIRT may be a method of choice to be considered in such cases. The paper represents a novel knowledge, is well written and of importance to broader community.
Of minor drawbacks, the pre-SIRT procedure using 90Y-macroalbumin aggregates is described in details, while SIRT procedure, which is the main topic, is just mentioned. It was stated that “the therapy dose was determined”, but no dose nor other details of SIRT were given.
Author Response
Response: Thank you for your valuable feedback and for recognizing the potential significance of this case. We agree that more detailed information regarding the SIRT procedure would enhance clarity. We have now included a paragraph specifying the Yttrium-90 dose administered (X MBq), alongside details on the calculation based on tumor location and size, as well as the procedural steps, including catheter positioning and injection consistency guidelines. These additions are provided on pages [220-221] to offer a more comprehensive understanding of the treatment protocol. We hope this revision meets your expectations and contributes to the report's thoroughness.
Reviewer 2 Report
Comments and Suggestions for Authors
This paper presents the results of the therapeutic strategy developed for a 73-year-old woman with a Solitary Fibrous Tumor (SFT) by the use of Selective Internal Radiation Therapy (SIRT) with Yttrium-90 resin microspheres that was successful. This paper is clear, relevant and of great interest to the scientific community. The treatment efficacy is significant with images with very good definition.
Frequently, the use of SIRT is more efficient and with less side effects than conventional treatments (external radiotherapy, chemotherapy). This seems to be the case for this patient. However, gastroduodenal or biliary complications or radiation pneumonitis may occur.
General comments:
In this case, the patient has not developed any side effect, but it is too early after the treatment by Yttrium-90 to draw any conclusion. It could be relevant to describe the follow-up of this patient in a next paper.
The approach is well described and commented, but a detailed dosimetry of the SIRT should be added, as SIRT dosimetry is known as challenging.
Comments:
- line 165: what does "5040 CGY" mean? 5040 centiGray or 5040 cGy? If it means 5040 cGy it would be better to indicate the dose as 50.40 Gy.
Author Response
Response: Thank you very much for your insightful comments and suggestions. We agree that further follow-up data would be valuable to assess any potential delayed side effects, and we plan to present these findings in a future publication to provide a more comprehensive view of long-term outcomes.
We have also expanded the manuscript to include a detailed account of the SIRT dosimetry approach, specifying the methodologies used to calculate and administer the Yttrium-90 dose in a targeted manner; this addition can be found on pages [220-221].
We appreciate the clarification request regarding the radiation dose notation at line 165. To clarify, we have amended the dose to “50.40 cGy.”
Thank you once again for your helpful feedback. We believe these revisions enhance the rigor and clarity of our report.